# p53 Family in Resistance to Targeted Therapy of Melanoma

**DOI:** 10.3390/ijms24010065

**Published:** 2022-12-21

**Authors:** Ignacija Vlašić, Anđela Horvat, Ana Tadijan, Neda Slade

**Affiliations:** Laboratory for Protein Dynamics, Division of Molecular Medicine, Ruđer Bošković Institute, 10000 Zagreb, Croatia

**Keywords:** melanoma, MAPK inhibitors, resistance, p53, p63, p73, p53 family isoforms

## Abstract

Metastatic melanoma is one of the most aggressive tumors, with frequent mutations affecting components of the MAPK pathway, mainly protein kinase BRAF. Despite promising initial response to BRAF inhibitors, melanoma progresses due to development of resistance. In addition to frequent reactivation of MAPK or activation of PI3K/AKT signaling pathways, recently, the p53 pathway has been shown to contribute to acquired resistance to targeted MAPK inhibitor therapy. Canonical tumor suppressor p53 is inactivated in melanoma by diverse mechanisms. The *TP53* gene and two other family members, *TP63* and *TP73*, encode numerous protein isoforms that exhibit diverse functions during tumorigenesis. The p53 family isoforms can be produced by usage of alternative promoters and/or splicing on the C- and N-terminus. Various p53 family isoforms are expressed in melanoma cell lines and tumor samples, and several of them have already shown to have specific functions in melanoma, affecting proliferation, survival, metastatic potential, invasion, migration, and response to therapy. Of special interest are p53 family isoforms with increased expression and direct involvement in acquired resistance to MAPK inhibitors in melanoma cells, implying that modulating their expression or targeting their functional pathways could be a potential therapeutic strategy to overcome resistance to MAPK inhibitors in melanoma.

## 1. Metastatic Melanoma—Progress, but Still no Cure

Metastatic melanoma is the most aggressive type of skin cancer and is responsible for the majority of skin cancer related deaths; its incidence has increased in the developed world in the last decades [1]. Melanoma has a high rate of somatic mutations compared to other solid tumors [2]. Frequent mutations occur in the mitogen-activated protein kinase (MAPK) pathway, including *BRAF*, *NRAS*, and *KRAS* genes. The *BRAF* gene encodes a serine/threonine protein kinase, which is an important regulator of the RAS/RAF/MEK/ERK kinase signaling pathway involved in many important cellular functions, including cellular proliferation, differentiation, and survival [3].

Somatic *BRAF* mutations have been found in nearly 60% of all melanoma, of which almost 90% harbor the V600E mutation, which results in the constitutive activation of MEK and ERK signaling, leading to increased cellular proliferation and survival and cancer progression [4,5]. The discovery of the BRAF V600E hotspot mutation led to development of targeted molecular therapies for melanoma [5]. Vemurafenib (PLX4032, Plexxikon or RG7204, Roche Pharmaceuticals), a potent inhibitor of BRAF (BRAFi) with high selectivity for BRAF V600E, was the first molecularly targeted therapy licensed for the treatment of advanced melanoma [6]. Initial response to vemurafenib was impressive compared to traditional chemotherapeutic agents; unfortunately, disease relapse was observed in patients within 6 to 8 months of therapy initiation [7,8]. Soon afterwards, combinations of specific inhibitors were shown to be more effective compared with single agent treatment [9,10,11]. Administration of combined BRAFi and MEK inhibitor (MEKi), i.e., vemurafenib with cobimetinib, or dabrafenib with trametinib, delayed acquired resistance and resulted in significantly improved progression-free survival (PFS; 13.7 months), overall response rate (ORR; 87%), and median overall survival (OS; 28.5 months) compared to BRAFi monotherapy [12,13]. Another approach in the treatment of metastatic melanoma that has emerged in the last decade is checkpoint inhibitor immunotherapy, aiming to promote elimination of tumor cells by immune response. Negative regulators of the immune response cytotoxic T-lymphocyte associated protein-4 (CTLA-4) and the programed death receptor 1 (PD-1), as well as its ligands, programmed death ligands 1 and 2 (PD-L1 and PD-L2), became main targets of tumor immunotherapies [14]. The expression of PD-L1 and 2 has been found in different cells, including melanomas [15]. The first approved immunotherapies used single antibodies (e.g., anti-PD-1 or anti-CTLA-4); however, the latest studies show significant increase in survival after using combinatory therapy, as compared to monotherapies [14,16]. Although combined immunotherapy is currently favored as the first therapy for metastatic melanoma, for advanced patients with unresectable and metastatic *BRAF*-mutated melanoma with high symptomatic disease burden, the combined BRAFi/MEKi targeted therapy remains the primary option due to its stronger/faster initial response [11,16]. Investigation of novel therapeutic approaches involving combinations of several anti-melanoma agents (such as dual targeting of BRAF/MEK and cyclin-dependent kinases CDK4/6) could provide significant improvement in the prognosis of *BRAF*-mutated metastatic melanoma patients [17,18].

## 2. Molecular Mechanisms of Resistance to Targeted Therapy

Generally, the main problem of effective cancer treatment is the rapid occurrence of resistance to drug therapy. Resistance to therapy may be intrinsic (pre-existing) or acquired (induced by treatment). A significant number, around 20%, of melanoma patients harboring BRAF V600E mutation show disease progression early after beginning targeted therapy treatment, indicating the presence of intrinsic resistance in a proportion of melanoma cells within the tumor. Drivers of intrinsic resistance include different oncogenic alterations (including PTEN or NF1 loss, *CCND1* amplification, *RAC1* or *HOXD8* mutations) or factors secreted by the tumor microenvironment (including HGF/c-MET and HIF-1α) [19,20]. Likewise, acquired resistance to BRAFi/MEKi involves various oncogenic mutations that can also cause reactivation of the MAPK (including *NRAS*, *KRAS*, and *MEK1/2* activating mutations, *BRAF* aberrant splicing, and *BRAF* amplification) and activation of the PI3K/AKT (*AKT1* mutation, loss of PTEN) pathway [21,22,23]. In addition to the mutations in the signalling pathways, other adaptive mechanisms like overexpression of MITF, persistent activation of receptor tyrosine kinases, expression of NGFR, nerve growth factor receptor (also known as CD271), and phenotype switching (a phenomenon of antagonism between proliferation and invasion driven by slow-cycling cell population) seem to have important roles in the development of therapy resistance in melanoma [24,25,26,27]. An unexpected role of the tumor suppressor p53 was recently discovered in a therapy-resistant melanoma subpopulation; p53 is stabilized by Wnt signaling, leading to the slow-cycling phenotype, one of the recently recognized hallmarks of BRAFi/MEKi therapy resistance [28]. Interestingly, BRAFi/MEKi treatment increases the level of NGFR (CD271) in drug-adapted, slowly-growing melanoma cell populations [26]. The NGFR is a crucial regulator of phenotype switching in melanoma, is important in controling melanoma cell growth vs. invasiveness [29], and positively controls gene networks associated with melanoma progression [30]. Furthermore, NGFR negatively regulates p53 pathway in melanoma-initiating cells and is required for different cell properties, e.g., stemness, proliferation, and tumorigenicity [31]. Thus, the targeting of NGFR induces apoptosis of BRAFi/MEKi-resistant melanoma cells and prevents melanoma invasion and metastasis formation in vivo [32]. Additional recurrent mutations and/or epigenetic changes of genes involved in PI3K/AKT signaling, cell cycle control (*RB1*, *CDKN2A*, and *TP53*), and other pathways were found in cutaneous melanoma; these regulate the course of the disease and could become targets of new therapeutic approaches [33,34]. The use of small molecules to inhibit the proteins involved in re-sensitization of melanoma cells harboring mutant BRAF to BRAFi/MEKi would enable future application of new drug combinations that could enhance sensitivity and/or delay resistance to targeted therapy in metastatic melanoma.

## 3. The p53 Family Isoforms

The tumor suppressor protein p53, classified as the “guardian of the genome”, elicits cell cycle arrest, apoptosis, and senescence in response to cellular stress, coordinating diverse signaling pathways. The p53 family comprises p53 itself, p73, and p63. Transcription from alternative promoters, alternative splicing, and diverse translation initiation sites contribute to the family complexity [35,36], and several protein isoforms with distinct N- and C- termini are encoded. Thus, all p53 family isoforms, apart from the full-length canonical p53 and TAp63α/TAp73α isoforms, are missing part of the N- and/or C-termini and, consequently, are deficient in some of the functional domains (Figure 1). Consequently, twelve p53 protein isoforms are encoded by a single *TP53* gene [37] and ten p63 proteins by the *TP63* gene; *TP73* can theoretically be transcribed into 35 different mRNAs, which could be translated into 28 different proteins, but so far not all of them have been found to be expressed in cell lines or tissues [38,39]. 

N-terminally truncated isoforms of p53 lack first 39, 132, or 159 amino acids and are called Δ40p53, Δ133p53, or Δ160p53, respectively. Consequently, Δ40p53 has lost TAD1, the first transactivating domain, but retains TAD2 and the entire DNA binding domain (DBD). ∆133p53 and ∆160p53 isoforms, produced from internal promoter P2, have lost both TADs as well as proline rich domain, PRD. Δ133p53 also lacks a part of the first conserved cysteine box of the DBD, which is completely deficient in Δ160p53, but both isoforms retain DBD. The p53 isoforms differ also in C-terminus. In contrast to α isoforms, which contain oligomerization domain (OD) and C-terminal domain (CTD), β and γ isoforms lack part of the OD and the entire CTD due to alternative splicing of exon 9 and premature termination codons (PTCs) [37,40]. 

There is a high degree of homology between the p53 and p63/p73 isoforms, the highest in DBDs, highlighting the role of transcription factors binding to the promoters of many overlapping target genes. The highest diversity between family members is at the C-terminus; instead of the CTD at the C-terminus of p53, the p63/p73 proteins possess a unique sterile α motif (SAM) domain, involved in protein–protein interactions and modulation of the transcriptional activity, as well as an inhibitory domain (ID). The SAM region of p63/p73 is subjected to intensive alternative splicing and, consequently, PTC. Only α isoforms contain an entire SAM domain. So far, five (α, β, γ, δ, and ε) and seven (α, β, γ, δ, ε, ζ, and η) different 3′-splice variants have been found for p63 and p73, respectively [38,41]. 

Both *TP63* and *TP73* genes generate two classes of isoforms, which are produced by alternative promoters and differ at N-termini: TAp63/TAp73 generated from P1 and containing the entire TAD, and those lacking it, ΔNp63/ΔNp73, transcribed from internal P2. The P1 transcript can be alternatively spliced, giving rise to other isoforms lacking the TAD (ΔEx2p73, ΔEx2/3p73, and ΔN’p73) [35,42,43,44]. Of importance, the ΔNp73 and ΔN’p73 transcripts encode the same protein due to the use of a second translational start site because of an upstream PTC in ΔN’p73 [42]. Accordingly, transactivating isoforms TAp63/TAp73 are potent transactivators of target genes and manifest tumor suppressor activities. In contrast, ΔNp63/ΔNp73 are mostly transcriptionally inactive and, in addition, they are dominant-negative inhibitors of p53 and TAp63/TAp73 isoforms. 

The transactivation activity of the isoforms of the p53 protein family is performed by the formation of tetramers. Two units form a dimer, which binds to a half-site on the consensus DNA sequence and is then stabilized by binding of the second dimer [45]. There is a significant cross-talk between family members in tumors, and the transcriptional activity of the individual tetramer depends on which isoforms it is composed of. Previously, we and other researchers determined the physical interactions between certain isoforms of p53/p63/p73, which can form heterotetramers that are involved in carcinogenesis [44,46,47,48]. The formation of mixed heterocomplexes between oncogenic (certain p53 mutants and p53 isoforms, ΔΝp63/ΔΝp73) and antioncogenic family members (wt p53, TAp63, and TAp73), which was confirmed under physiological conditions in mammalian cells, correlates with the loss of transactivation of their target genes and, consequently, the loss of suppressor functions [35]. The dominant-negative effect of the oncogenic isoforms is performed either through heterocomplex formation or through competition for promoter binding with p53, TAp63, and TAp73 [49,50].

The expression and stability of p53 family isoforms can be modulated by several mechanisms on different levels (transcriptional, posttranscriptional, translational, and posttranslational), influencing their biological activities and functions (extensively reviewed in [39]).

Co-expression of different p53 isoforms and their potential interactions contribute to the diverse biological activities and functions of p53. Accordingly, their unbalanced expression can cause cancer, premature aging, inflammation, developmental disorders, or deficiency in tissue regeneration [37,40]. Many biological functions of p53 isoforms have been described, including cell-cycle regulation, apoptosis, senescence, DNA repair, stem-cell regulation, cancer stemness, metabolism, autophagy, cellular invasion, migration, and angiogenesis, immunosuppression, and inflammation (reviewed in [39]).

The diversity in structure leads to diversity in subcellular localization and consequently in various biochemical/biological activities, which are cell-type dependent. Finally, p53-mediated cell response is the sum of the activities of co-expressed p53 isoforms [37]. Currently, the roles of p53 isoforms in tumor formation are still being investigated. The p53 isoforms cannot be categorized as exclusively oncogenic or tumor-suppressive since their biological activities and thus their prognostic values are associated with the cell context.

While the mutations of *TP53* are frequent in human cancers, the mutations of *TP63* are not common in somatic cells, and *TP73* essentially is never targeted by inactivating mutations [35].

## 4. The Role of p53 Family Isoforms in Melanoma 

In melanoma, p53 and other family members, p63 and p73, fail to function as tumor suppressors and to regulate target genes related to apoptosis and cell cycle, implying that deviant functioning of p53 could support melanoma progression [51]. Reduced levels of p53 or its mutations contribute to aggressiveness and resistance to therapy [52,53,54,55]. *TP53* and *TP63* genes are mutated in 27 and 22% of melanoma samples, respectively, according to cBioPortal [56,57,58] (Figure 2). Several diverse mechanisms of p53 inactivation in melanoma have been proposed, including mutations of *CDKN2A* (*CDKN2A* encodes for both p16INK4A and p14ARF), upregulation of MDM2 (an E3 ubiquitin ligase that controls p53 expression and function) or MDM4 (negative regulator of p53) overexpression, activation of iASPP (inhibitor of apoptosis stimulating protein of p53) or deubiquitinase USP5, and silencing of the *TP53* gene by epigenetic mechanisms [55,59,60,61,62]. However, this phenomenon is not fully understood.

### 4.1. The Expression and Activities of p53 Isoforms in Melanoma 

A handful of clinical studies reported the expression of p53 isoforms in several tumor types, confirming that small molecular weight p53 isoforms might play an important role in tumorigenesis [63,64,65,66,67,68,69,70]. In addition, a paucity of studies have investigated the expression and biological functions of p53 family isoforms in melanoma (Figure 3, [28,69,71,72,73,74,75,76,77,78,79,80]). It has been shown that the human melanoma cell lines express a broad pattern of p53 isoforms, including p53α, p53β, ∆40p53α, ∆133p53α, ∆133p53β, and ∆160p53α, with the latter being the most variable. Interestingly, their expression differed from primary melanocytes. Δ160p53α, and to a minor degree, Δ160p53β, can be recruited to chromatin, and all ∆160p53 isoforms are shown to stimulate proliferation and in vitro migration [79]. The ∆160p53 isoforms are shown to bear pro-oncogenic traits, since they contribute to mutant p53-induced properties, e.g., increased survival, proliferation, migration, adhesion, and invasion, and thus are required for pro-oncogenic “gain of function” p53 [81]. In addition to ∆160p53α, elevated levels of small molecular weight p53 isoform ∆133p53α were observed in metastatic melanoma tumors compared to normal tissues [73]. It has been shown that all Δ133p53 isoforms promote invasion, with Δ133p53β being the most efficient. The overexpression of Δ133p53β promotes cancer stem cell potential and metastasis and correlates with a worse cancer patient outcome, including for melanoma [73,82,83]. Additionally, Δ133p53 has been shown to promote invasion and metastasis of B16 melanoma cells to the lungs, dependent on secreted factors, including IL-6 and the chemokine CCL2 [74]. Additionally, the elevated levels of Δ133p53β isoform promote an immunosuppressive environment in prostate cancer by regulating the expression of *CD274,* which encodes PD-L1 [84] and boosts a chemoresistant environment in glioblastoma [85], leading to aggressive cancer. Therefore, along with ∆160p53, Δ133p53 acts in a similar manner to the “gain of function” mutant p53 proteins to promote migration, invasion, and metastasis, which may contribute to poor survival in patients with Δ133p53-expressing tumors. Recent findings have shown that the Δ133p53β activity is negatively regulated through aggregation. However, its interacting partners, such as CCT chaperon complex or ΔNp63, a p53 family isoform, can recruit Δ133p53β from aggregates, thus contributing to its tumor invasive activity [86]. Similar to Δ133p53, Δ40p53 can exhibit the dominant-negative effect on p53α and can alter p53-mediated transcriptional activity, apoptosis, and growth suppression when co-transfected with p53. It has been shown that both Δ40p53 and p53β have increased expression in melanoma cell lines compared to fibroblasts and melanocytes and show aberrant subcellular localization, and their expression can be induced by DNA damaging agents, e.g., cisplatin. Interestingly, these two isoforms can alter p53 function in melanoma cells; Δ40p53 can inhibit while p53β can enhance the p53-dependent transcription of p53 target genes, *p21* and *PUMA* [69]. Similarly, Δ40p53, independently of full-length p53, promotes cell survival by activating the transcription of the antiapoptotic ligand netrin-1. Inhibiting netrin-1 causes apoptosis and inhibits tumor growth in vivo; a positive correlation was found between Δ40p53 and *netrin-1* gene expression in human melanoma biopsies. Interestingly, knockout of FLp53 by sgRNA increased the expression of Δ40p53 in human skeletal myoblasts [87]. Though, when exogenously overexpressed, Δ40p53 can increase the level and activate endogenous p53, and can promote apoptosis over cell-cycle arrest (even with γ-irradiation) in melanoma cells and thus reactivate p53-dependent tumor suppression function and impact melanoma cell fate [76]. In melanoma tissues, Δ40p53β expression was reduced compared to healthy tissue, while reduced p53β expression or increased p53α mRNA expression correlated with poorer overall survival of melanoma patients [73].

### 4.2. The Expression and Activities of p73 Isoforms in Melanoma

In contrast to p53 and p63, p73 is essentially never mutated in cancer, but it is often overexpressed [35,88], including in melanoma [89]. Increased expression of p73 in metastatic melanoma could imply that the p73 is a positive regulator of melanoma progression from primary tumor to metastasis [89]. There is a paucity of studies that have analyzed the p73 isoforms’ expression in melanoma. A study of the expression and effect of particular p73 isoforms in metastatic melanoma showed overexpression of TAp73, Ex2p73, and Ex2/3p73 (spliced transcripts derived from the first promoter), whereas ΔNp73 was the predominant isoform in benign nevi [72], which is in line with our findings of gene expression [73]. On the protein level, decreased expression of ΔNp73β but increased expression of ΔNp73α was observed in metastatic melanoma tissue compared to healthy tissue [73]. Significantly greater expression of ΔNp73α protein in melanoma [73] is reasonable considering that N-terminally truncated isoforms, e.g., ΔEx2/3p73, can be expressed more in melanoma metastasis compared to primary melanoma. Furthermore, ΔEx2/3p73 was shown to drive EMT-like (epithelial-to-mesenchymal transition, EMT) phenotypic switch, migration, and invasion of melanoma cells via EPLIN depletion and IGF1R-AKT/STAT3 signaling. These changes can be reversed with TAp73, confirming once again the interplay between p73 and N-terminally truncated isoforms. In vivo, ΔEx2/3p73 expressing tumors were significantly more invasive and developed micrometastases in lungs and liver. ΔEx2/3p73 levels positively, while EPLIN negatively, correlates with Breslow depth of the primary melanomas, and higher level of ΔEx2/3p73 transcript in combination with a loss of EPLIN expression was found in melanoma metastases compared to the primary group [90]. Furthermore, it has been shown that ΔNp73 overexpression enhances tumor vascularization and increases the angiogenic potential of B16-F10 melanoma cells in vivo. In more detail, higher vessel density indicated by the number of CD31+ structures, higher mitotic index, and increased expression of VEGF-A were observed in B16-ΔNp73 tumors a few weeks after injection into C57BL/6 mice, which supports the pro-angiogenic role of ΔNp73 in melanoma cells [75]. In addition to ΔNp73β, the expression of TAp73α and TAp73β was reported in a panel of human melanoma cells, and significantly higher expression of TAp73 was observed in wild-type cell lines for both BRAF and p53 compared to other mutation groups [79]. This is in line with the results of previous findings, where expression of both proapoptotic TAp73 and anti-apoptotic N-truncated p73 isoforms were observed in melanoma cell lines, suggesting that their ratio could also determine potential drug response in melanoma. Indeed, upregulation of TAp73β expression by adenoviral transfection enhances the sensitivity of melanoma cells to standard chemoterapeutic agents, such as adriamycin and cisplatin, both in vitro and in vivo [71]. Since TAp73α is associated with the suppression of apoptosis in non-melanoma cancer cells [91], the abovementioned findings also imply that the treatment-mediated apoptosis depends on the content of p73 C-terminus in a cell context/type–dependent/specific manner. Protein interactions between TAp73α, TAp73β, ΔNp73α, and p53β with the p53α protein were observed in the A375M melanoma cell line (Hanžić et al., unpublished results), some using already reported FRET-FLIM analysis [92], implying that these isoforms could be involved in altering p53 function in melanoma. 

### 4.3. The Expression and Activities of p63 Isoforms in Melanoma

The third member of the p53 family, transcription factor p63, is recognized as an important regulator of the development of stratified epithelia, including skin. A complex network of p63 transcriptional targets involves both positive and negative regulation, which orchestrate processes crucial for the development and differentiation of the skin [93]. p63 can repress the expression of *CDKN1A,* which encodes the cyclin-dependent kinase inhibitor p21, and *HES1,* which is an effector of the Notch pathway, thus maintaining cell proliferation in basal keratinocytes (basal layer of the epidermis) [94,95]. In addition, p63 activates the expression of several genes important for cell adhesion (*ITGA6* and *ITGB4* encoding integrins, *BPAG1* and *PERP* encoding components of hemidesmosomes, *CDH3* encoding P-cadherin, *FRAS1* encoding extracellular protein, and *KRT14* encoding for the component of keratin intermediate filaments) [96,97,98,99,100,101]. The ΔNp63α isoform was shown to be the most frequently expressed among the p63 isoforms in normal skin and cutaneous tumors [93]. p63 is frequently expressed in undifferentiated and poorly differentiated tumors that originate from epithelial cells [102]. There remains controversy regarding the p63 expression in melanoma. Initial studies rarely revealed p63 expression in malignant melanoma [102,103,104]; however, more recent studies report the existence of p63 expression and mutations in cutaneous melanoma [77,105]. p63 was found to interact with p53 in melanoma, thereby influencing its tumor suppressor role. It seems that p63 has an oncogenic role in melanoma, since increased expression of p63 on a gene and protein level was observed in melanoma cell lines and clinical tumor samples. In addition, its reactivity correlates with worse clinical outcome of melanoma patients [77]. Furthermore, p63 seems to be a negative regulator of apoptosis through a twofold mechanism in melanoma, e.g., translocation to the mitochondria, subsequently influencing expression of BCL-2 family members and repression of p53 in the nucleus. By acting as a dominant-negative inhibitor of p53, p63 renders melanoma cells resistant to standard chemotherapy and targeted BRAFi therapy [77]. 

## 5. The Role of p53 Family Isoforms in Resistance to Targeted Therapy

The mutations of *TP53* and *TP63* are found in more than 20% and 15% of the BRAF-mutated tumors, respectively (Figure 2), and are certainly involved in the acquisition of resistance. Nevertheless, there is increasing evidence indicating the influence of specific p53 family isoforms in acquired resistance to MAPK inhibitor (MAPKi) targeted therapy. For example, the specific isoform expression pattern, such as increased potentially pro-oncogenic Δ40p53β isoforms and reduced tumor-suppressive TAp73β isoforms, was detected in both primary and metastatic melanoma cells with acquired resistance to BRAFi targeted therapy, i.e., vemurafenib. In addition, reduced Δ133p53β expression was observed in BRAFi-resistant melanoma cells. Furthermore, reduced levels of TAp73 and ΔNp73 were observed in BRAFi-resistant primary melanoma cells with the activation of the PI3K/AKT pathway, while increased levels of TAp73 and ΔNp73 were detected in resistant metastatic melanoma cells with the re-activation of the MAPK pathway [79]. The BRAFi-resistant primary melanoma cells show features of slow-cycling cells, e.g., mesenchymal morphology, reduced proliferation and migration, increased resistance to chemotherapeutic agents, i.e., cisplatin and etoposide, as well as altered cell cycle profile and levels of cell cycle regulators [106], implying that the specific p53 isoform expression pattern could correlate with specific features of BRAFi-resistant melanoma cells. Interestingly, the slow-cycling cell phenotype, known to be a feature of a targeted therapy-resistant melanoma cells [28,106], can be prevented by the inhibition of p53, thus sensitizing melanoma cells to BRAFi/MEKi-targeted therapy [28]. In more detail, it has been shown that the slow-cycling phenotype is driven by non-canonical Wnt signaling via the Wnt5A protein, which stabilizes and utilizes p53. Wnt5A promotes expression of p53 and p21, which drive cells into a slow-cycling state where they resist therapy. Furthermore, a single dose of pifithrin-α, a p53 inhibitor, sensitized melanoma cells in vivo and in vitro to BRAFi/MEKi, while the BRAFi/MEKi therapy increased the number of p53-expressing cells in melanoma tissue [28]. Although inhibition of p53 could have disadvantages, the possibility of abolishing MAPKi resistance in slow-cycling cells dependent on the Wnt5A/p53 axis by using a single dose of p53 inhibitors could be a potential therapeutic strategy to overcome MAPKi resistance.

Interestingly, melanoma cells with acquired resistance to MAPKi, i.e., BRAFi (vemurafenib) or combined with MEKi (trametinib), can show reduced levels of TAp73 and enhanced sensitivity to platinum-based agents [80]. In more detail, TAp73 was shown to mediate resistance toward platinum-based agents and influence DNA damage response by regulating the nucleotide excision repair (NER) mechanism in melanoma cells. Consequently, lower TAp73 levels reduce the efficiency of NER and enhance the accumulation of DNA double-strand breaks after treatment with platinum-based agents in MAPKi-resistant melanoma cells. These results provide the possibility of stratifying patients with MAPKi-resistant melanoma, dependent on TAp73 expression status, which could benefit platinum-based chemotherapy [80].

As mentioned, by interacting with p53 and influencing its tumor suppressor function, p63 acts as a negative regulator of apoptosis and contributes to chemoresistance. It has been shown that the treatment with BRAFi or chemotherapeutic agents, e.g., etoposide, paclitaxel, and cisplatin, increases the apoptosis of melanoma cells upon depletion of p63. In addition, the depletion of p63 caused increased expression of phosphorylated ERK1/2 and MEK, thereby most probably elevating the activity of the MAPK pathway in the A375M melanoma cell line, which was more pronounced upon BRAFi treatment. This implies that the reactivation of the MAPK pathway, as a mechanism of melanoma resistance to BRAFi therapy, could be regulated via the p63 pathway [77]. In addition, overexpression of ΔNp63β was shown to elevate EGFR expression, resulting in increased expression of phosphorylated MEK1/2 in WM164 metastatic melanoma cells. These results further support the role of p63 in influencing MAPK signaling via EGFR in melanoma cells [78]. Interestingly, melanoma cells with acquired resistance to MAPKi, i.e., BRAFi (vemurafenib) or MEKi (trametinib), show increased expression of both TAp63 and ΔNp63 isoforms on an mRNA level as well as ΔNp63α/β/γ on a protein level in different melanoma cell lines. In addition, the upregulation of p63 was shown to contribute resistance to targeted MAPKi therapy in cell lines and clinical samples, and by modulating its expression, MAPKi-resistant melanoma cells can be sensitized to MAPKi treatment. Interestingly, increased p63 level was shown to be a result of lower degradation, dependent on E3 ubiquitin ligase FBXW7, which is negatively regulated by MDM2 [78], a known negative regulator of certain p53 and p73 isoforms [107,108,109,110,111,112]. Consequently, in MAPKi-resistant melanoma cells, nuclear enrichment of MDM2 most probably resulted in downregulation of FBXW7 and subsequent upregulation of p63. Therefore, the targeting of MDM2 by inhibitor Nutlin-3A re-established/upregulated FBXW7 in a p53-dependent manner and resulted in p63 degradation and thus made MAPKi-resistant melanoma cells susceptible to MAPKi treatment [78]. These results suggest the possibility of abolishing MAPKi resistance dependent on the MDM2/FBXW7/p63 axis by using Nutlin-3 as a therapeutic strategy to defeat MAPKi resistance. 

In addition to their potential involvement in response to BRAFi/MEKi targeted therapy, the regulation and activity of the p53 protein family members should also be taken into account in other novel approaches to melanoma treatment. It was recently found that CDK4/6 inhibitors (e.g., palbociclib), which are currently being tested clinically as a potential therapy for melanoma, can activate p53 through modulation of alternative splicing of MDM4, which is a known negative regulator of p53. This regulation is mediated by the suppression of PRMT5, a protein arginine methyltransferase, an epigenetic modifier that, among other functions, modulates pre-mRNA splicing affecting also MDM4, leading to lower levels of MDM4 expression and consequently to increased p53 expression. Disruption of the palbociclib activity on the MDM4-PRMT5 pathway is one of the hallmarks of the development of drug resistance to CDK4/6 inhibitor therapy. However, a potent and prolonged response to the CDK4/6 inhibitor was achieved by combining with the PRMT5 inhibitor, leading to suppression of cell proliferation and tumor growth [18]. Locus *CDKN2A* is the most frequently affected gene by germ-line mutations in cutaneous melanoma. It encodes two distinct tumor suppressors, namely p16INK4A and p14ARF, which positively regulate retinoblastoma (RB) and p53 pivotal tumor suppressors (by inactivating MDM2), respectively [113]. A novel study on early stage primary cutaneous melanoma shows significant presence of somatic mutations in *TP53* and *CDKN2A*, being present in 26 and 16% of the analyzed samples, respectively [114]. This and other studies continue to emphasize that the role of p53 tumor suppressor has to be considered during melanoma development and response to therapy. Furthermore, the development of novel drugs targeting alternative splicing processes could be used as a novel therapeutic approach in melanoma [115].

## 6. Conclusions

Metastatic melanoma is one of the most aggressive tumor types, with frequent mutations mostly affecting components of the MAPK pathway, such as kinase BRAF. Regardless of positive initial response to MAPK inhibitors, disease relapse occurs due to acquired resistance largely as a result of reactivation of MAPK or activation of PI3K/AKT pathways. The diverse biological functions and overall activity of p53 and its family members, p63 and p73, are a result of the balance between different p53 family isoforms. Most of the p53 family isoforms interact with each other to form heterotetramers that interfere with transactivation ability or assemble inactive homotetramers that compete for DNA binding. Thereby, proteins with the transactivation domain can imitate the function of p53, transactivating many p53 target genes, whereas proteins without it show a dominant-negative effect toward p53 and its family members. The p63 was found to interact with p53 in melanoma, thereby influencing its tumor suppressor role. *TP53* and *TP63* mutations occur in a respectable number of the *BRAF*-mutated melanomas, while p73 is never mutated but overexpressed. Moreover, in melanoma there is unbalanced expression of different p53 family isoforms, which are also shown to contribute to melanoma aggressiveness and to influence response to therapy. In this review, we summarized the p53 family isoforms that have imbalanced expression in melanoma and their potential to modulate p53 function, exhibiting specific biological functions. Furthermore, we emphasized the findings showing the involvement of specific p53 family isoforms in acquired resistance to MAPK inhibitor targeted therapy. Therefore, revealing the specific expression patterns and roles of the p53 family isoforms could potentially lead to uncovering of novel therapeutic targets in melanoma. In addition, modulating expression of the p53 family isoforms or targeting their functional pathways linked to MAPK inhibitor resistance could be a potential therapeutic strategy to overcome resistance to MAPK inhibitors in melanoma.

## Figures and Tables

**Figure 1 ijms-24-00065-f001:**
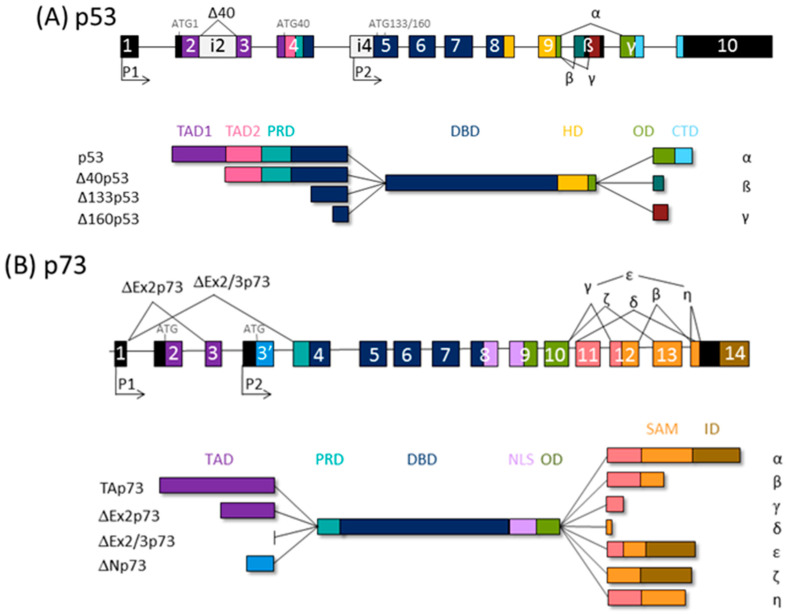
The p53 family gene architecture and generation of the protein isoforms. The scheme of the (**A**) human *TP53,* (**B**) *TP73,* and (**C**) *TP63* gene (upper) and protein (lower panels). TAD, transactivation domain; PRD, proline-rich domain; DBD, DNA binding domain; ID, inhibitory domain; HD, hinge domain; NLS, nuclear localization signal: OD, oligomerization domain; CTD, C-terminal domain; SAM, sterile alpha motif; ID, inhibitory domain (model adapted from [39]).

**Figure 2 ijms-24-00065-f002:**
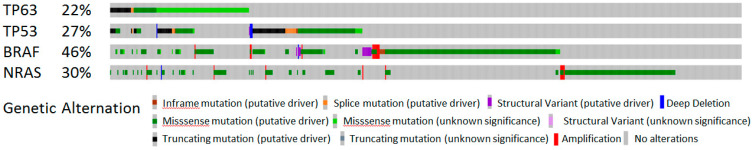
Genetic alterations of *TP53*, *TP63*, *BRAF,* and *NRAS* in 696 melanoma patients/samples (downloaded data from cBioPortal Oncoprint view, https://www.cbioportal.org/, accessed on 27 October 2022) [56,57,58].

**Figure 3 ijms-24-00065-f003:**
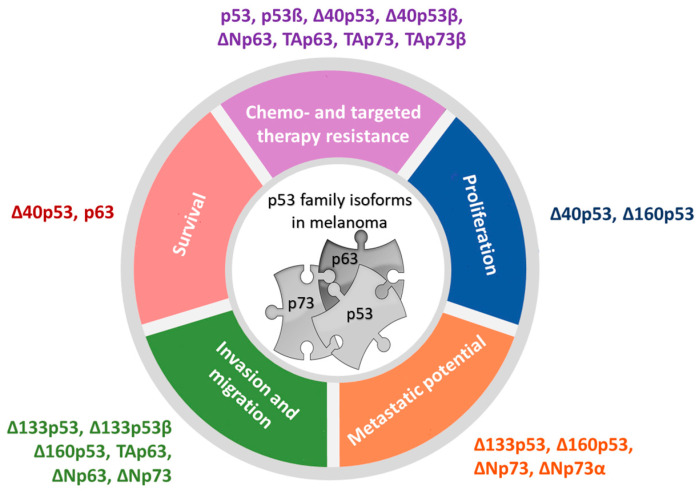
Biological functions of p53 family isoforms in melanoma (model adapted from [39]).

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
