# Peer review of "p53 Family in Resistance to Targeted Therapy of Melanoma"

_ijms, 2022, doi:10.3390/ijms24010065_

Round 1

Reviewer 1 Report

X the authors

The review by Vlasic et al describes the role of p53 family isoforms in melanoma and propose their regulation as a potential therapeutic strategy to circumvent melanoma resistance to MAPK inhibitors.

The authors provide an exhaustive description of p53 isoforms with a map of genetic alterations of p53 and p63 reported in more than 20% of melanoma samples (cohorts of 696 melanoma patients), which may open the way for the studies of their roles and the discovery of novel therapeutic targets.

The ms is well outlined, conceived, and clearly represented.

I only have few suggestions to improve the flow of the article:

-       Enlarge characters of fig 3

-       Split chapter 4 and chapter 5 with subparagraph regarding p73 and p63

Reviewer 2 Report

A review by Vlasic et al focuses on the role of p53 in resistance of melanoma cells to therapy. The aim of the review is interesting as targeted therapy did not show expected clinical benefits. The Authors should include more recently published papers. In addition, the focus on the relationship between p53 and CD271 in melanoma should be more emphasized. Several important papers in the fields have been missed, e.g., PMC7371866 and PMC6531451

Specific comments:

1. line12: "BRAF inhibitor targeted therapy" please change to "BRAF inhibitors".

2. Fig. 1 - the Authors show almost identical panel as is published in another paper (ref. 34). 

Round 2

Reviewer 2 Report

All comments have been addressed.